# Psychology of the Embrace: How Body Rhythms Communicate the Need to Indulge or Separate

**DOI:** 10.3390/bs7040080

**Published:** 2017-11-29

**Authors:** Sabine C. Koch, Helena Rautner

**Affiliations:** 1Research Institute for Creative Arts Therapies (RIArT), Alanus University Alfter, 53347 Alfter, Germany; 2Department of Therapy Sciences, SRH University Heidelberg, 69123 Heidelberg, Germany; 3Department of Psychology, University of Heidelberg, 69117 Heidelberg, Germany; helena.rautner@stud.uni-heidelberg.de

**Keywords:** interpersonal resonance, embodiment, movement rhythms, haptic communication, movement qualities, movement analysis, Kestenberg Movement Profile (KMP), rhythmicity, musicality, vitality affects

## Abstract

In the context of embodiment research, there has been a growing interest in phenomena of interpersonal resonance. Given that haptic communication is particularly under-researched, we focused on the phenomenon of embracing. When we embrace a dear friend to say good-bye at the end of a great evening, we typically first employ smooth and yielding movements with round transitions between muscular tensing and relaxing (*smooth*, *indulging rhythms*), and when the embrace is getting too long, we start to use slight patting (*sharp*, *fighting rhythms* with sharp transitions) on the back or the shoulders of the partner in order to indicate that we now want to end the embrace. On the ground of interpersonal resonance, most persons (per-sonare, *latin* = to sound through) understand these implicit nonverbal signals, expressed in haptic tension-flow changes, and will react accordingly. To experimentally test the hypothesis that smooth, indulgent rhythms signal the wish to continue, and sharp, fighting rhythms signal the wish to separate from an embrace, we randomly assigned 64 participants, all students at the University of Heidelberg, to two differently sequenced embrace conditions: (a) with the fighting rhythm at the end of the sequence of two indulgent rhythms (Sequence A: smooth-smooth-sharp); and (b) with the fighting rhythm between two indulgent rhythms (Sequence B: smooth-sharp-smooth). Participants were embraced for 30 s by a female confe­derate with their eyes blindfolded to focus on haptic and kinesthetic cues without being distracted by visual cues. They were instructed to let go of a handkerchief that they held between the fingers of their dominant hand during the embrace, when they felt that the embracer signaled the wish to finish the embrace. Participants significantly more often dropped the handkerchief in the phase of the fighting rhythm, no matter in which location it occurred in the embrace sequence. We assume that we learn such rhythmic behaviors and their meaning from the beginning of life in the communication with caregivers and meaningful others. Some are universal and some are quite idiosyncratic. Infants seem to be highly sensitive to the dynamic nuances presented to them, demonstrating a high capacity for embodied resonance and a high behavioral plasticity. Such adaptive mechanisms are assumed to lay the foundations of family culture (including the degree to which nonverbal cues are attended to, the communication of taboos, etc.) and larger culture, and may also play an important role in interpersonal attraction and aesthetic experience.

## 1. Introduction

Implicit nonverbal signals serve to communicate from early infancy onward. A child cries in distress and the mother picks him/her up and moves him/her up and down on her knees, bouncing him/her in a sharp rhythm. He/she slowly starts to calm down and breathes more regularly, while the mother gradually changes the rhythm to a soothing low-intensity swaying rhythm (smooth rhythms; for operationalization see Appendix A; and directly from 1.4 on) that softly rocks him to rest. We all use movement rhythms to address the needs of others, express ourselves, and communicate with others. In early infancy, they are almost our only form of expressing our needs and thus have survival value.

In an extended good-bye embrace, the movement sequence just described reverses: first, when the wish to indulge is predominant, we can observe smooth indulgent rhythms, then—when the embrace starts to get too long for one of the embracers—we can observe sharp fighting rhythms from his/her part indicating the wish to separate. This sequencing of movement rhythms in embraces was described by the psychiatrist, child psychoanalyst, and movement specialist Dr. Judith S. Kestenberg (Kestenberg & Sossin, 1973/1979; Kestenberg, 1995) [1,2]. Kestenberg systema­tically analyzed basic movement dimensions, such as the movement rhythms, along with their meaning; observed their developmental sequence; built a theory about developmental stages; and explored and revised this theory with a group of other researchers and practitioners experientially (movement-based). She related this theory to pathology and health, particularly mental health and disorders of the embodied self (Kestenberg Amighi, Loman, Lewis, & Sossin, 1999) [3].

Movement rhythms are the tension-flow changes that occur in the body, when our muscles go from tense to relaxed and vice versa: the constant change of tension in our body. They build a continuous stream of information for us about where and how our body is, what we need, and how we do (e.g., whether we are “in danger” or “safe”; Bruner & Postman, 1947) [4]. Movement rhythms indicate needs and serve a perceptive (interoceptive/proprioceptive/exteroceptive), expressive, and communicative function (Koch & Sossin, 2012) [5].

### 1.1. Bidirectionality of Movement and Affective-Cognitive System

As early as 1872, Charles Darwin in his book “The expression of emotions in men and animals” [6] provided many examples of humans and animals using movement to express themselves and communicate with conspecifics and other species. While Darwin focused mainly on the *expressive function*, almost exactly 100 years later, psychology started to pay investigative attention to the *bidirectionality of movement and the affective-cognitive system*, investigating the *impressive function* in the so called facial- and body feedback hypotheses (LaFrance, 1985; Laird, 1980; Riskind, 1984; Wallbott, 1990) [7,8,9,10]. The impressive function of movement on affect and cognition was subsequently further investigated by looking at how facial expressions (Havas, Glenberg, Gutowski, Lucarelli, & Davidson, 2010 [11]) and postures (Rossberg-Gempton & Poole, 1992 [12]) as well as gestures (Cacioppo, Priester & Berntson, 1993 [13]) changed emotion, cognition, and even action (Bargh, Chen, & Burrows, 1996 [14]). Books in phenomenology and cognitive sciences, as well as robotics, picked up upon the principles of embodiment and the effects of the body on the mind (Gallagher, 2006; Gibbs, 2005; Pfeifer & Bongard, 2006 [15,16,17]). The traditional Cartesian split was strongly questioned (Damasio, 1994; Merleau-Ponty, 1962; Sheets-Johnstone, 1999 [18,19,20]), and the plasticity of the body–mind system came more and more to the foreground (Pert, 1997; Keysers, 2011 [21,22]).

### 1.2. Dynamic and Interactive Effects of Body Feedback: Including Kinaesthetic and Haptic Communication

Yet, the body feedback research focused on static gestures or postures and did not employ any dynamic movement. Taking into account the perspective of movement therapy and movement analysis, we conducted a number of experiments to demonstrate the effects of dynamic movements on affect and cognition, with a particular focus on the influence of movement qualities (Koch, 2011; 2014) [23,24]. Cacioppo et al. (1993) [13] had demonstrated that approach movements caused positive affect and attitudes to formerly neutral objects (Chinese ideographs), whereas avoidance movements caused negative affect and attitudes toward the same objects. Building upon their findings, we varied approach and avoidance movements in smooth or sharp movement rhythms (four conditions) to find out that only in the smooth rhythm condition did the formerly known effects of approach and avoidance motor behavior (Cacioppo et al., 1993) [13] on affect and attitudes occur (Koch, 2014 [24]). That is, movement quality (here movement rhythms) had a decisive influence on the effects of movement shape changes on affect and attitudes. Affect and attitudes were more positive after an approach movement than after an avoidance movement (confirming the findings of Cacioppo et al., 1994) [13], while no difference occurred when a fighting rhythm was employed with either an approach or an avoidance movement (Koch, 2014) [24]. The fighting rhythm seemed to overwrite the meaning of the approach and avoidance difference, possibly because, with its onset, the organism was getting into a defensive state (having to defend against all incoming divergent information) to focus on a specific task to be completed. This is actually quite straightforward and functional: the fighting or indulgent movement quality indicates tension or relaxation, danger or comfort (Bruner & Postman, 1947) [4], and our attentional system reacts accordingly.

Yet, the interactive meaning of body rhythms remained to be demonstrated. Social embodiment research is needed to follow up with interpersonal body feedback experiments. A first study was conducted by Koch, Berger, and Schorr (reported in 1.6.1 below, and Koch, 2011) [23], focusing on the *communicative meaning of rhythms in handshakes*, where results suggested indulgent rhythms indicate and cause positive affect and signal agreeableness, openness, and extraversion (from the Big 5 personality dimensions), whereas fighting rhythms indicated negative affect and neuroticism.

There is much differentiated knowledge on movement, its logic, its basic dimensions, and movement analysis in the field of dance movement therapy (e.g., Bartenieff & Lewis, 1980; Kestenberg, 1995; Laban, 1980; Lamb, 1965) [2,25,26,27]. Embodiment research could profit a lot from making use of that knowledge in theory, methods, and applications. Particularly, theory is strong in movement analysis (Kennedy, 2010) [28], and can help to advance the knowledge about embodiment more rapidly and integratively than the mostly isolated effects that have been empirically tested so far. If empirically tested effects are related to theory systems of movement analysis, missing information on aspects that require further attention will start to stand out. The research could then generally be conducted more systematically, informed by causal hypotheses from already existing experientially and phenomenologically validated systems. Movement analysis systems in turn could be further validated or revised in aspects where quantitative empirical tests yield according results. 

### 1.3. Primacy of Movement and Expression: Roots of Rhythms Research in “Ausdruckspsychologie”

The baby moves already before birth and, by moving (use of proprioception) and touching, develops the first experiences of separating interoception (perception of signals from the inside of the body) from exteroception (perception of signals from outside of the body), differentiating a first sense of “me” and “not me” via surface sensitivity (Sheets-Johnstone, 1999) [20] and recognition of a membrane that separates the inside from the outside (which may be our first act of self-consciousness; Sheets-Johnstone, 1999) [20]. This membrane is permeable and prone to rhythmic stimulation. What it comes down to in any form of rhythmicity/musicality (Trevarthen, 1999) [29] is our ability to resonate using our body, and the biggest sensory organ our skin as a membrane that can bring exteroception and interoception in unison. Proprioception, sense of balance, and kinaesthesia have been widely underestimated as a sensorimotor basis of affect and cognition (Schönhammer, 2013; Allesch, 2009) [30,31]. Body rhythms provide *structure*, patterns, contrast, figure and ground, temporal sequencing, etc. We can perceive and create rhythms with our bodies. Creating rhythms provides a person with an experience of structure and control. This can be very important in clinical contexts for patients with depression, anxiety disorders, traumatic experiences, schizophrenia, and other mental health problems (Fuchs & Koch, 2014) [32], because rhythm can provide experiences of agency and self-efficacy, both crucial components of psychological health (Bandura, 1977) [33].

As early as in the 1920s, the term *rhythm* was used in the German “Ausdruckspsychologie” (Kirchhoff, 1965) [34] to describe movements, and the terms *smooth/sharp* (“rund/eckig”) were used to describe the transition between two movements (Oseretzky, 1929) [35]. At the same time, an experiment by Gestalt psychologist Wolfgang Köhler (1929) [36] showed that the neutral (non-existing) words *bouba* and *kiki* would be significantly more often assigned to either a smooth or a sharp geometric form. *Bouba* was consistently assigned to the round shape (that was similar to an ink plot or a puddle), and *kiki* was assigned to the sharp form (that was similar to a star shape). Poffenberger and Barrows (1924) [37] discovered that curved and smooth lines (round shapes) were significantly more assigned to states of mind such as sad, quiet, or gay, whereas sharp lines were significantly more assigned to arousal or rage. The *aesthetic aspect* of these shapes was investigated by Bar und Neta (2006) [38], who demonstrated that smooth, curved objects on picture cards (such as sketched vases, sofas, etc.) would be evaluated as significantly more positive and beautiful as the same objects in a sharp lined format. Wallbott (1982) [39] and Aronoff, Woyke and Hymann (1992) [40] were the first European researchers who investigated movement qualities experimentally. Both built on the theories of Laban and experimental findings from cognitive psychology and emotion research. Aronoff et al. (1992) [40] showed that smooth and sharp movement affected emotions in distinct ways: Smooth and round facial features and movement qualities in the sense of smooth or flowing transitions would be rated more positively than sharp facial features and movements with abrupt transitions, which were experienced as threatening. These findings yield empirical evidence that smooth transitions in a range of modalities such as facial features, sound, and rhythm can cause other feelings and visceral sensations than sharp transitions (see Kestenberg, 1995; Fuchs & DeJaegher, 2009) [2,41]. The studies of Wallbott (1982) [39] and Aronoff et al. (1992) [40] were not yet making the connection to body feedback research, and likewise they failed to differentiate shape and quality in an unequivocal way. Yet, they laid important foundations for further research. Koch (2014) [24] started to experimentally separate shape and quality changes in movement, demonstrating their independent influence on affect and attitude change. By relying on Kestenberg Movement Profile-theory (KMP-theory), the present embrace study again separates both aspects, allowing to investigate the dimension of movement quality (smooth vs. sharp rhythms) independently of the dimension of movement shape regarding their affective and communicative effects. The effects of the dynamic movement quality on attitudes and affect has already been discussed above in the studies on *dynamic body feedback* (Koch, 2011) [23].

### 1.4. Basic Movement Dimensions: Tension-Flow Rhythms and Shape-Flow Rhythms in the KMP

In the 1970s, movement analyst Judith S. Kestenberg developed a theory system of movement and meaning that she related to developmental processes and clinical meaning: the Kestenberg Movement Profile (KMP; Kestenberg & Sossin, 1973; Kestenberg, 1975 [1,2]). She based the system on two theories: Laban Movement Analysis (Laban, 1980) [26] and psychoanalytic developmental theory (A. Freud, 1992/1937) [42], and subdivided movement into movement qualities (the tension-flow system) and movement form (the shape-flow system). Kestenberg hypothesized among other elements *ten movement rhythms* that express different *needs* of a person and can help to understand needs of preverbal infants and their communication with the caretaker. These movement rhythms fall into two important categories: *indulgent* (with smooth transitions between single rhythms) and *fighting* (with sharp transitions or reversals between single rhythms). Kestenberg exemplified in her teaching that when we embrace a friend, we usually first indulge in rhythms that are smooth, and then use tapping or patting on the back of the partner in order to indicate the wish to separate (Kestenberg, 1995; Loman & Kestenberg-Amighi, 1996, personal communication, 28 January 1996) [1].

#### 1.4.1. Tension-Flow Rhythms

Rhythms are small (micro-)movements that accompany us throughout life (while they are universal in meaning, they differ in attributes from person to person and thus reflect an important part of our personality). There are ten basic or pure rhythms. The first developmental rhythm is the *sucking rhythm*; it is hard-wired as a reflex and occurs already in the womb (Loman, 1992) [43]. The newborn baby is increasingly able to use it more consciously to help organize its entire body. This rhythm serves as self-soothing and is often still seen in adults who try to calm themselves down in stressful situations. The next developmental rhythm is the *biting rhythm*, which is the first “fighting” rhythm, that is, one with sharp reversals. One can see it, for example, in lectures, when people start to tap with one of their feet, or bite on a pencil in order to focus or concentrate. The *biting rhythm* serves for separation, categorization, and selection. Both of these “oral” rhythms develop in the first year of life. All fighting rhythms serve as transition into the next phase (Kestenberg, 1995; see also Erikson, 1963) [2,44]. What follows are *twisting, straining* (second developmental year), *running-drifting, starting-stopping* (third year), *swaying, birthing* (fourth year), *jumping*, and *spurting-ramming rhythms* (fifth year; all rhythms see Figure 1; for more background on meaning and indicated needs, see Kestenberg-Amighi et al., 1999; Koch & Sossin, 2012) [3,5]. In each phase, an indulgent rhythm is followed by a fighting rhythm (one for indulging into and one for separating out of each developmental phase); each yield/relaxation is followed by a separation/tension (Erikson, 1963) [44].

The rhythms are notated by handwriting on a piece of paper on a timeline. The notator’s body serves as a “seismograph” to capture the (continuous) tension changes in the body of the observed person with her own kinaesthetic empathy (to be transmitted into the writing arm) (Fischman, 2008; Koch & Sossin, 2012) [5,46]. This method may appear subjective, however, rhythms ratings from a variety of empirical contexts have consistently yielded inter-rater-reliabilities of *ICC*’s higher than 0.80, making it suited for use in research (Koch, Goodill & Cruz, 2002; Koch, Martin, Schubert, Fuchs, & Mombaur, 2017) [47,48].

#### 1.4.2. Shape-Flow Rhythms

Shape-flow rhythms are the continuous changes in growing and shrinking movement of the body. The prototype of growing is inhaling and of shrinking is exhaling. Growing can be observed in the body when an individual is comfortable in his/her environment or with the persons present. Growing and shrinking toward a certain stimulus is approach and avoidance motor behavior. A small child may be shrinking away from the barking dog and growing toward mummy. From growing and shrinking movements, we judge comfort or discomfort of preverbal infants or animals in our surroundings. All organisms on our planet seem to be tied to this principle: when the stimulus is evaluated as “good/nourishing” (e.g., food, persons, sunshine) the organism grows toward it, when it is evaluated as “bad/harmful,” it shrinks away from it (even plants follow this rule; Koch, 2011) [23]. Preceding this evaluation, there needs to be a rudimentary form of self-recognition: any animal needs to recognize any part of his/her body as his/her own to know the rule: “when hungry, don’t eat yourself” (Dennett, 2001) [49]. Back to shape-flow changes, the breast-fed baby will turn toward the mother’s body when hungry, and turn away from the breast when full; the regularly sensitive, “good enough” mother (Winnicott, 1960) [50] will recognize the meaning of these shape-flow utterances and react accordingly.

Growing and shrinking can occur multi-directionally (bipolar shape-flow), that is, growing in all directions, for example, when suddenly coming into a sunny and warm environment; and it can also occur uni-directionally (unipolar shape-flow), when reacting to one particular stimulus (such as food, or a person in the room; Kestenberg-Amighi et al., 1999) [3]. Shape-flow movements are thus growing and shrinking movements in reaction to the environment: they indicate comfort or discomfort (bipolar shape-flow: growing or shrinking related to the general environment), or attraction or repulsion (unipolar shape-flow: growing or shrinking related to one particular stimulus, termed *approach or avoidance motor behavior* in cognitive science) in a situation.

### 1.5. From Individual to Interpersonal Body Feedback: An Embodied Approach to Embraces

On the basis of social embodiment theory (Niedenthal, 2007) [51], and particularly the findings on the bidirectionality of movement and cognitive-affective system (Neumann & Strack, 2000; Zajonc & Markus, 1984) [52,53] in static as well as dynamic movements (Koch, 2014) [24], we planned the present study. We aimed to carry this line of research on the effects of dynamic body movement one step further into interpersonal research on body feedback to explore the interactive meaning of body rhythms. As we had seen from the experiments of Koch (2014) [24], movement qualities in the form of smooth and sharp rhythms had an affective component with sharp rhythm causing more negative affect, but also causing a defensive reaction when it came to rudimentary attitudes: sharp movement qualities overwrote the effect of approach and avoidance motor behavior on attitudes; the effect of approach and avoidance motor behavior on attitudes/evaluation was only retained when the rhythm transitions enacted were smooth. In embraces, this effect of individual and natural approach and avoidance is now extended to include interpersonal body feedback, and has a clear cultural component in addition. 

### 1.6. The Influence of Tactile Feedback in Interpersonal Communication 

In all studies described here, effects of movement have been observed on the affective or the evaluative measures (rudimentary attitudes), not on the cognitive ones. All studies so far had only included the proprioceptive-kinesthetic feedback on one’s own affect and attitudes (individual level). To go one step further, we included the interactive processes and consequences of dynamic motor feedback. In this interactive study, we focused on the haptic channel of communication and investigated the communicative implications of movement rhythms.

Touch is a domain rarely investigated in the social sciences (for an exception see Schubert, Waldzus, & Seibt, 2008; Seibt, Neumann, Nussinson, & Strack, 2008) [54,55]. Montagu (1971) [56] had formulated how essential and central touch and skin contact are for the development of a human being. He stated that “the personal identity has only insofar substance and structure, as it has its fundament in the reality of our bodily experiences” (Montagu, 1971) [56]. The sense of touch can already be observed in eight-week-old fetuses retreating after a touch of the lips. At this point in time, the embryo is only 2½ centimeters tall and has neither eyes nor ears. In agreement with Montagu (1971) [56] on the role of tactile communication, Fuchs (2000; p. 114; first author’s translation) [57] states that “the reciprocity of the relation is in no other sense modality as pronounced as in the sphere of touch. Visual auditory and the other ‘distance senses’ use mediating means such as light and air. The skin is both the separating and the connecting interface, both sense organ (impression) and active communication organ (expression)”.

We worked with embraces as an independent variable, because they are a relevant social phenomenon of nonverbal communication. In Germany, people do not frequently embrace or kiss in public context, and the main body contact in greetings and saying good-bye is the handshake (see Koch, 2011) [23]. Thus, embraces in Germany are for close friends, family, and lovers, more so than in neighboring countries such as France or the Netherlands, where one greets with two or three light kisses on the cheek (which also implies a closer social distance, where you can not only sense but also smell the person). For a more differentiated understanding of cultural implications of touch see (Burgoon, Guerrero, & Floyd, 2016; Gallace & Spence, 2010; McDaniel & Andersen, 1998) [58,59,60]. An embrace always has an active and a passive component: to touch (tactile) and to be touched (haptic), which is only actually a shift in focus, since any touch is always “ambiguous” in this sense (Merleau-Ponty, 1962) [19]. Since our participants were more on the “being hugged/touched” side in this case, we mainly speak of haptic communication.

In this study, we varied the tension-flow rhythms of the embraces to include one fighting rhythm among two indulgent rhythms, keeping the shape-flow constant. There were *two relevant preliminary studies* preceding this study: An experimental study on the effects from dynamic body feedback from handshakes (Koch, Berger, & Schorr; reported in Koch, 2011) [23], and a naturalistic study on embraces in the field (Koch, Skibka, Steiner, & Grassinger, 2011) [61].

#### 1.6.1 Preliminary Study 1: What do Handshakes Communicate? (Koch, Berger, & Schorr; in Koch, 2011) [23] 

Because we were not content with the fact that social embodiment research did not include the interactive aspect of rhythms communication, we set up interpersonal body feedback experiments. The first one was carried out by Koch, Berger, and Schorr (reported in Koch, 2011) [23], and focused on the communicative meaning of rhythms in handshakes, employing two fighting rhythms vs. two indulgent rhythms with 64 participants. One confederate was trained to shake the hands of the participants in two different sequences: (a) containing two different indulgent rhythms (á 20 s; sucking and swaying) or (b) containing two different fighting rhythms (á 20 s; biting and straining). Results suggested that rhythms have a direct influence on the perception of affect and personality of the handshaker: round rhythms indicated to our participants that the person was in a better mood, more agreeable, open, and extraverted, whereas sharp rhythms indicated to them that the person was in a worse mood and more neuroticistic. Round rhythms furthermore influenced the mood of the participant in a more positive direction than sharp rhythms (Koch, 2011) [23].

#### 1.6.2 Preliminary Study 2: Embraces in Natural Contexts (Koch, Skibka, Steiner, & Grassinger, 2011) [61]

Before starting the embrace experiment, we first observed how embraces were carried out in natural contexts (Koch, Skibka, Steiner, & Grassinger, 2011) [61]: Would we be able to see the two phases assumed by Kestenberg? In a field study on two campus cafeterias of major University cities and at two train stations in the same two cities in Southern Germany, we observed *N* = 123 pairs of persons, embracing each other, either to greet or to say good-bye. The observed sequences in the field study provided a relatively clear picture that Kestenberg’s assumptions were grounded in the observable reality of free embraces in middle Europe of 2010. The rhythm of the main part of the observed embrace was smooth in 83.5% of the cases, and sharp in 16.5%. The rhythm of the separation part was a bit less clear in its pattern, with 49.5% smooth and 50.5% sharp rhythms observed. The *patterns of rhythm sequencing* was: smooth-smooth (46.6%), smooth-sharp (36.9%), and sharp-sharp (15.5%), and thus confirmed the sequencing assumed by Kestenberg (sharp-smooth only occurred in 1.0% of the persons observed). However, we would have expected less round rhythms toward the end of the embrace. The sharp-sharp pattern was observed exclusively in men embracing other men: in Germany, it seems to be culturally indicated to start patting right away in same-sex embraces between men, probably to not let any questions about sexual orientation come up (i.e., homophobia still seems to impede indulgence between men in public places). In sum, it is important to note that none of the observing students was a KMP-notator, but they were merely trained to observe the rhythms in a 30 min training, therefore observational results need to be interpreted with the according caution (Koch, Skibka, Steiner, & Grassinger, 2011) [61]. Moreover, standing at a distance would sometimes impede observation of micro-rhythms and tension-drops, and are thus another potential source of error.

#### 1.6.3 Aim of this Study

The aim of the present study was to carry dynamic embodiment effects from body rhythms further into interactional resonance between persons, using observations of micro rhythms and their phases in an embrace; our study thus focused on the interactive aspects of kinesthetic/haptic body feedback effects. It may contribute to detect mechanisms of interpersonal resonance and further our understanding of communication of *primary intersubjectivity* (Trevarthen, 1979) [62], *embodied affectivity* (Fuchs & Koch, 2014) [32], and *affective intentions* (Shai & Belsky, 2011a; 2011b) [63,64], with prototypes stemming from early infant–caregiver interaction (Stern, 1985; Trevarthen & Delafield-Butt, 2013) [65,66]. The study further aims to synergize knowledge traditions from cognitive science (e.g., Gibbs, 2005) [16] and movement analysis (e.g., Kestenberg, 1995) [2] to contribute to a more informed embodied cognition and movement analysis research.

### 1.7. Hypothesis and Operationalizations

We tested whether the smooth vs. sharp movement rhythms serve as communicative signals to continue with or end an embrace (haptic semantics), with the hypothesis that changes in rhythmic movement qualities from indulgent (smooth) to fighting (sharp) are used in interpersonal communication to indicate the wish to separate; assuming that the haptic signal for the wish to continue or end an embrace will be understood from the change in rhythmic movement qualities the participants are exposed to. Following KMP-theory, smooth rhythms (tension-flow changes with smooth transitions) were assumed to indicate the wish to continue with or indulge in the embrace, while sharp rhythms (tension-flow changes with sharp transitions) were assumed to indicate the wish to separate. This was measured by employing a signal (drop of the handkerchief they held in their hand) that the participants were supposed to provide, when they experienced or noticed that the person embracing them wanted to separate. We chose this method because we did not want the embracer to get irritated or reactive, or the sequences of the embrace to get disturbed by any reaction of the person being embraced (non-reactive method). The drop of the handkerchief was in most cases not noticed by the embracer, in the remaining cases not noticed for sure (from the notes of the embracer). It was, thus, the best non-reactive method we could come up with. Regarding the outcomes, we assumed that the handkerchief would more often be dropped after the onset of the sharp movement rhythm during the embrace. The outcome (=time of signal) was measured in milliseconds (ms) from the onset of the embrace to the time the handkerchief was dropped by the participant. The times were then also related to the rhythm phases. In our experiment, only tension-flow was varied in order to isolate the effect of movement qualities (smooth vs. sharp rhythms), that is, the person embraced tried not to provide any shape-flow cues (such as approach or avoidance behavior signals with the torso), keeping the same distance during the entire embrace.

Under the assumption that there are sex differences in nonverbal sensitivity that may influence the results (Hall, 1990) [67], with women usually scoring higher on nonverbal sensitivity, we selected the sex of participant as a control variable. Because the Need for Interpersonal Touch (NFIPT; Nuszbaum, Voss, Klauer, & Betsch, 2010) [68] may exert a similar influence, with persons scoring higher displaying a higher nonverbal sensitivity, we selected the NFIPT-score as a second control variable.

## 2. Methods

### 2.1. Study Design

In a one-factorial between-group design, with the *movement sequence* as the independent variable and the *time of signal* (duration from onset of the embrace to the point in time of dropping the handkerchief) as the dependent variable, and *sex* as well as *the score on the Need for Interpersonal Touch Scale* (NFIPT, Nuszbaum et al., 2010) [68] as control variables, we conducted a group comparison to detect whether the sharp, fighting rhythm was causal for the dropping of the handkerchief. The drop was requested at the point in time the participants noticed that the person embracing them wanted to end the embrace. The sequences of the movement rhythms were as follows (see Figure 2):

The between-subject factor *movement sequence* with two levels consisted of Sequence A (sm-sm-sh) and Sequence B (sm-sh-sm). This solution with two experimental conditions controlling each other, with *N* = 61 participants, reached the necessary power (0.88) according to computations with g*power (Erdfelder, Faul, & Buchner, 1996) [69], on an α-level of 0.05 (Bortz & Döring, 1995, S. 579) [70], assuming a medium effect size (Cohen, 1992) [71].

### 2.2. Sample

The sample consisted of 61 participants (37 women; and 24 men). The mean age was *M* = 21.66 (*SD* = 2.56), with an age range from 18 to 29 years. Twenty women and 12 men were in Condition 1 (Sequence 1), and 17 women and 12 men were in Condition 2 (Sequence 2). The conditions controlled each other, we did not employ any other control condition. Participants were randomly assigned to the experimental conditions yet matched by gender (i.e., random numbers were applied, but the men were filled up evenly into each of the two groups). Participants were recruited on the central business street of a Southern German University town, and via bill boarding in the Psychological Institute. All participants were students. The majority of the sample consisted of psychology students (*n* = 47, i.e., 77%), the rest of the participants had other study majors (*n* = 14, i.e., 23%). One participant needed to be excluded, because he dropped the handkerchief beyond the 30th second on both occasions (see Section 2.4.1 below). There were no prerequisites to fulfill for participation, since the study was about intuitive perception of one’s own affect and the subtle nonverbal commu­nication cues. For this reason, there was no limitation in age or cultural background of the participants. The only requirement was fluency in German to insure complete understanding of the questionnaires. Participants received course credits (30 min) or 5 Euros after the completion of the procedure.

### 2.3. Procedure

Participants were welcomed at the experimental room by a female experimenter and provided informed consent before the start of the experiment. Then, they completed the MBAS-scale (movement-based affect scale of the KMP; Koch & Müller, 2007; Koch, 2014) [24,72] and were asked to read the *instructions for the embrace* (Box 1). The experimenter informed them that a female person concealed behind the folding screen would embrace them, and that they would be blindfolded to not be distracted by any visual cues. After making sure that all instructions were understood and no questions remained, the experiment started. After a brief relaxation (three deep inhales), the first embrace was given (test run). The embracer then either gave Sequence A or Sequence B of the embrace in silence, keeping the shape-flow constant (the shape of the embrace was always: opening, enclosing, application of the three movement rhythms; at the end of the sequence: opening and stepping back). The person giving the embrace was self-cued for the phase changes, reading the 3 × 10 s from a large wall clock, she was directly looking at. Each participant received two identical embraces (test run and trial run), from the female confederate that had been trained in the application of KMP-rhythms. The time was stopped by the experimenter starting at the onset of the embrace and ending with the time, when the participants dropped their handkerchief. After the second embrace, participants filled in the MBAS-scale again and indicated the experienced closeness and naturalness of the embrace. Then, the Need for Interpersonal Touch Scale (NFIPT-Scale; Nuszbaum et al., 2010) [68] was administered. Finally, the participants answered questions in an open format to capture their experience and observa­tions, and for manipulation check. A test of the experimental set-up had shown that for participants, it was important to know the sex of the embracing person. The according information on the gender of the embracer was thus given in the beginning. After the embrace, the embracer provided an estimation of the perceived muscle tone of each participant, a feature potentially helpful for identifying defensive states (usually, the higher the more defensive). After the debriefing, participants received their course credits and/or the 5 Euros and had the opportunity to see the person who had embraced them. Almost everybody took advantage of this offer. The experiment took 30 min.

Box 1Embrace-Instructions for the participants, given in writing.**Instructions**You will now receive an embrace by a female person. Take your time to get ready.- The embrace will last 30 s, meaning it is a bit longer than a regular embrace.- You may possibly experience changes during the embrace.- If you feel like it, you can join into the embrace (i.e., respond to the embrace); the most important thing, however, is that you are able to focus on the embrace; when you get the impression that the person embracing you wants to end the embrace, please let go of the handkerchief that you have received and hold in your hand, let it drop to the floor.- There is no right or wrong point in time, simply follow your bodily sense.- The study is not about, when *you* want to end the embrace, but when you get the impression that *the person embracing you* wants to end the embrace.Now, please take three deep breaths and relax. Whenever you are ready to receive the embrace, let the experimenter know.

### 2.4. Materials & Instruments

The study took place in a 50 m^2^ room on the ground floor of the back building of the Psychological Institute at the University of Heidelberg, Germany. The embracer used a regular wall clock (indicating the seconds) to keep track of the time switches between the embrace phases and a stop-watch (type) for measuring the dropping of the handkerchief from the onset of the embrace operated by the experimenter. Demographic data was assessed as part of the questionnaires prior to and after the embrace. 

#### 2.4.1. Time of Signal (Duration up to the Drop of the Handkerchief)

The time of signal (our main dependent measure) was operationalized as the duration from the onset of the embrace to the signal (drop of the handkerchief) by the participant, measured in seconds by a regular stop-watch. The length of the embrace was 30 s, each phase lasting 10 s (as described above; see Section 2.1 and Section 2.3).

In total, nine participants dropped the handkerchief only after the 30 s in the main trial (second embrace, using the *shape change* or the stepping back of the embracer as a cue, rather than the *rhythm change*). In order to not lose these persons from the data set, we took the values from the preliminary trial (first embrace) from eight out of those nine participants, in which all of the eight had dropped the handkerchief in the interval between 11 and 19 s. This way, only one person needed to be excluded from data analysis: the one who dropped the handkerchief after the entire 30 s were over in both trials. Because the phases were identical in both embraces, we decided to proceed this way. Participants probably just waited for the shape-flow signal of the embracer the second time (i.e., the opening of the arms and the one step back). While this was also ‘correct’ (i.e., a valid signal indicating that the embracer wanted to end the embrace), it was not the point of our study: we were interested in whether they would notice the tension-flow signal, that is, the rhythm change to a fighting quality.

However, to be on the safe side with the computational solution, we also computed the infe­rential statistics for the complete first trial excluding the ones that had waited until the 30th s (*n* = 7), and then computed the inferential statistics for the complete second trial excluding the ones that had waited until the 30th s (*n* = 9), and found very similar same *p*-values and effect sizes in all three ways of computing the main results (see Appendix B for an overview). We thus decided to employ the computational solution described above, where all but one participants were kept in the data set (*N* = 60) for the main analysis.

#### 2.4.2. Affect Measure (MBAS-Questionnaire; Koch, 2014) [24]

The German and English Version of the KMP-Questionnaire (Koch, 1999) [73] evolved into the Brief KMP-based Affect Questionnaire (Koch & Müller, 2007) [72], later renamed to Movement-Based Affect Scale (MBAS; Koch, 2014), and followed the theory and movement analysis method of Kestenberg (Kestenberg & Sossin, 1973/1979) [1] described in the introduction (Section 1.4) employing a lexical analysis of the movement and meaning connections from the book “The meaning of movement” (Kestenberg Amighi et al., 1999) [3]. The original scale resulted from the extraction of interpretations (interpretative terms) of movement and meaning in the main tension-flow and the shape-flow system/categories and encompassed 64 items (Koch, 1999) [73]. The internal consistency was then assessed with a sample of *N* = 80 students. The scale yielded very good reliabilities with *ICCs* > 0.80 for all items. The KMP-Questionnaire (Koch, 1999) [73] thus laid the grounds for the reflection of rhythms in their communicative, semantic, diagnostic, and therapeutic function. Koch and Müller (2007) [72] created a psychometrically sound short version, the Movement-Based Affect Scale (MBAS), which has already been applied in several studies. The MBAS (Koch & Müller, 2007; Koch, 2014) [24,72] is a 13-item instrument measuring the movement-based affect of a person following KMP-theory (see Appendix C). It contains eight items of tension-flow such as “tense vs. relaxed”, “held back vs. playful, coy”, and “yielding vs. fighting”, and five items of shape-flow such as “open vs. closed” and “approaching vs. avoiding”, on a seven-point bipolar scale connecting the two poles (Appendix C). Analysis of internal consistency yielded very high values with *Cronbach’s* α = 0.92.

#### 2.4.3. Experienced Naturalness of the Embrace (Stimmigkeitsskala; Rautner, 2012) [74]

Participants rated the experienced naturalness of the embrace on a bipolar seven-point scale consisting of the following six items: The embrace was: “authentic vs. inauthentic”, “fitting vs. non-fitting”, “true vs. untrue”, “natural vs. unnatural”, “regular vs. strange”, and “comfortable vs. uncomfortable”. All items were then inversely poled, in order to have higher numbers indicate higher naturalness. The internal consistency of the scale was very good with *Cronbach’s* α = 0.90 after the first embrace, and *Cronbach’s* α = 0.93 after the second embrace.

#### 2.4.4. Need-for-Interpersonal-Touch-Scale (NFIPT; Nuszbaum et al., 2010; 2014) [68,75] 

The Need for Interpersonal Touch Scale (NFIPT; Nuszbaum, Voss, Klauer, & Betsch, 2010) assesses individual differences in preference for touch in general, and the preference for using touch as an additional source of information specifically. The construct was originally derived from the construct “Need for Touch” (NFT) from consumers’ research, which postulates that touching objects changes consumer behavior. Touch of a product increases the affect and attitude toward the object, and the experienced sympathy, evaluation, and satisfaction with the product selection. Nuszbaum, Voss, Klauer, and Betsch (2010) [68] found such effects only when the NFT was high. Beyond consumers’ research, the construct can be transferred to the interpersonal realm (Erceau & Guéguen, 2007; Fisher, Rytting & Hesslin, 1976; Gallace & Spence, 2009) [76,77,78] to phenomena such as interpersonal touch in interaction situations. The German scale was developed at the Universities of Freiburg and Heidelberg, and validated with *N* = 60 participants, assessing the frequency of touch initiated by the participant, which significantly correlated with the NFIPT-score (*r* = 0.32). Higher frequency of touch of the participant led to a higher satisfaction with the evaluation of the partner, the experimental setting, and one’s own evaluations, but only for persons with a high NFIPT-score. 

The NFIPT-scale (Nuszbaum et al., 2010) [68] consists of 24 items that measure the need for interpersonal touch. The sum value indicates the individual trait with values >0 indicating a high vs. values of < 0 indicating a low need for interpersonal touch. The scale ranges between the poles of −3 (= applies not at all) to +3 (= applies completely/totally), with the mean at 0. Sample items are: “It can happen in a conversation that I touch my conversation partner”, “When communicating via smartphone or internet, I miss the bodily closeness”, “Interpersonal touch strengthens confidence”, or “I generally avoid bodily contact with other people”, “I don’t like others touching me in a conversation”. In this study, we tested whether a higher NFIPT-score causes (a better detection of the signal or) a more positive affect after the embrace. Reliability analysis of internal consistency yielded very high values with *Cronbach’s* α = 0.87, and no outlying items. 

#### 2.4.5. Other Variables

The embracer furthermore rated the *muscle tone* of the participant on a three-point scale (high-regular-low). We wanted to be able to control our assumption that persons displaying a particularly high muscle tone may not be as sensitive (or permeable enough) for the rhythms or rhythmic signals as others. High tension can be assumed to impact personal and interpersonal resonance (Koch, 2014) [24]. 

#### 2.4.6. Manipulation-Check 

For manipulation check, we used an open question “How was the embrace for you? Did you notice anything?” The hypotheses of the participants regarding the aim of the study were checked for systematic biases and all hints were noted (e.g., reluctance to participate).

### 2.5. Statistical Analysis

We first computed a *Chi*^2^-test to see *in which phase of the embrace the signal was given most frequently* and whether that difference was statistically significant. Following this, we computed an analysis of covariance (ANCOVA) to investigate the group differences between the two sequences of embraces controlling for gender and NFIPT-score. We used SPSS 18 with the alpha level set at 0.05.

## 3. Results

### 3.1. Descriptive Statistics

The descriptive statistics are provided in Table 1. 

### 3.2. Inferential Statistics

#### 3.2.1. Phase of Signal: In Which Phase Was the Handkerchief Most Frequently Dropped?

In order to determine the frequency of the signal for the single phases (see Table 2 crosstab), we computed a 2 × 2 *Chi*^2^-test (one-sided). The time measure was split into three phases of 10 s each (according to the three rhythms employed) and the participants dropping time was categorized accordingly, into phase 1, 2 or 3 (or 4, if it lay beyond the 30 s).

The means of Sequence A with *n* = 30 participants was 23.63 (SD = 4.35), the means of Sequence B with *n* = 29 participants was 18.09 (SD = 6.05). The frequencies are provided in Table 2. 

In the condition of Sequence A, there were 24 persons who dropped their handkerchief in the correct phase (Phase 3), as opposed to eight persons who dropped it in an incorrect phase: one in Phase 1, six in Phase 2, and one in Phase 4. In the condition of Sequence B, there were 22 persons who dropped their handkerchief in the correct phase (Phase 2), as opposed to seven persons who dropped it in an incorrect phase (Phase 3). The *Chi*^2^-test was significant on the 1%-level with Pearson-*X*^2^(59) = 18.45, *p* = 0.000017. For the main analysis, we then computed an analysis of covariance.

#### 3.2.2. Main Analysis

To test the main hypothesis, we computed an ANCOVA with rhythm sequence (A: sm-sm-sh vs. B: sm-sh-sm) as the independent variable, the signal (i.e., the time from the onset of the embrace to the drop of the handkerchief), and the affect change (post values minus pre-values of the MBAS) as the dependent variables, and sex and the NFIPT-score as the control variables. There were no outliers for the variable *time of signal*. Effect sizes below are reported from the between-group effects for the single variables tested (univariate effects). 

*Time of signal.* There was a significant difference in time of signal (onset of embrace to drop of handkerchief) between the participants in Sequence A and Sequence B: the participants in Sequence B dropped the handkerchief significantly earlier F(1, 60) = 18.90, *p* = 0.000, eta2 = 0.25 (*Cohen’s d*: 1.15; using DeCoster’s effect size converter at www.stat-help.com) than the participants in Sequence A (see Table 3 and Figure 3).

*Changes in affect.* There was no significant difference (no difference whatsoever) in participants’ affect change (measured with the MBAS; Koch & Müller, 2007) [72] from before the embrace to after the embrace *F*(1, 60) = 0.007, *p* = 0.934, *eta*^2^ = 0.00 (*Cohen’s d:* 0.10; using DeCoster’s effect size converter at www.stat-help.com) (see Table 3).

*Control variables.* There was no significant gender effect of the person embraced *F*(1, 60) = 0.027, *p* = 0.869; *eta*^2^ = 0.00 (*Cohen’s d*: 0.13). There was no significant influence of the control variables *sex* on time of signal *F*(1, 60) = 0.018, *p* = 0.894, *eta*^2^ = 0.00 (*Cohen’s d*: 0.12), or affect change *F*(1, 60) = 0.506, *p* = 0.480, *eta*^2^ = 0.00 (Cohen’s d: 0.19), and *Need for Interpersonal Touch (NFIPT-score)* on time of signal *F*(1, 60) = 0.240, *p* = 0.626, *eta*^2^ = 0.00 (*Cohen’s d*: 0.17), or affect change *F*(1, 60) = 0.159, *p* = 0.692, *eta*^2^ = 0.00 (*Cohen’s d:* 0.15; for all *Cohen’s d*s using DeCoster’s effect size converter at www.stat-help.com).

*Other variables.* We did not find any significant correlation of muscle tone with the outcomes, *r*(1; 59) = 0.09, *p* = 0.495 for *time of signal*, and *r*(1; 59) = 0.13, *p* = 0.337 for affect change. There was no significant differences in muscle tone between male and female participants. Further, perceived naturalness, perceived closeness, and affect after the main embrace were positively correlated on the 1%-level. 

### 3.3. Manipulation Check 

Five out of 61 participants stated explicitly that they perceived the ‘patting’ (sharp rhythm) as the signal to end the embrace, yet one of the five said that the patting was not strong enough to let go of the handkerchief. All others (*n* = 56) were unaware of any specific signal, confirming the implicit, non-deliberate character of rhythm communication. 

In response to the question “*How was the embrace?*” participants enumerated the following attributes (single participants’ reactions separated by a semicolon): “Comforting; artificial/strange/unnatural in this setting, but not uncomfortable; distant and rehearsed; mechanical; had three/different phases (5) (with different meanings, 1); usually an embrace has rather one phase/way; unnatural, too long/longer than a natural embrace; massaging, relaxing; I relaxed into it; uncomfortable/unnatural; fine; tender and comfortable; the more rapid the patting, the more the feeling that the other wanted to separate; tense; very comfortable, comforting, calming; forced; comfortable, easy; a bit artificial; with a growing tension as time continued; comforting; comfortable and caring; staged, rehearsed; artificial but still warm; phases seemed to contradict each other in meaning; very comfortable up to the patting; irritating; too intimate; warm and honest; unnatural and strange; I got the feeling the person wanted to comfort me; a bit artificial, but still comfortable; transmitting friendship and affection; unnatural and strange, because of the long duration; staged; some phases were familiar others not; even though I dislike situations like this, more comfortable than expected; schematic; distanced, cautious, an unusual massage; different phases; standardized; protective quality, but still stiff and unnatural, since we did not know each other”.

## 4. Discussion

### 4.1. Effects of Movement Rhythms in Embraces

In the experimental trial at hand, we examined the effects of smooth vs sharp movement rhythms in an embrace on participants’ perception of the wish to continue or to end the embrace. Relying on the bidirectionality assumption from *embodiment* research (Neumann & Strack, 2000; including body feedback findings from rhythms, Koch, 2014) [24,52] as well as Kestenberg’s theory on movement rhythms and their meaning (Kestenberg & Sossin, 1973/1979; Kestenberg Amighi et al., 1999) [3,5], we hypothesized that for the participants, a rhythm with sharp transitions would signal the wish to separate, whereas a rhythm with smooth transitions would signal the wish to continue the embrace. Participants held a handkerchief in their dominant hand during the embrace by a female confederate, and were instructed to simply let go of it once they perceived that the embracer wanted to separate. Participants dropped their handkerchief significantly more often in the phase with the sharp rhythm transitions than in any other phase of the embrace. There were no affect changes from before the embrace to after the embrace. Participants did not show more positive affect after having received the hug, which may partly be attributed to the experienced artificiality of the laboratory situation, and more generally to the wide variety in which the embrace was experienced by the participants, from strange and uncomfortable to comfortable and comforting (see enumeration in manipulation check, Section 3.3). There was no influence of sex of the embraced person, or of need for interpersonal touch (operationalized with the NFIPT-score) on the outcomes. The findings of the manipulation check indicated that only five persons (8%) of our participants were aware of the specific signal communicating the wish to separate (i.e., the change from a smooth to a sharp movement rhythm), but that 78% of the participants clearly understood the message of the signal, that is, for at least 70% of the participants who correctly understood the signal, this understanding was not tied to awareness of the communicative cue. The findings, thus, confirm the assumption that tension-flow-rhythms mostly operate on an unconscious or implicit level as part of tacit communication.

Our findings suggest that smooth vs. sharp movement rhythms have a clear semantic of their own. Separation is facilitated by movement qualities with sharp transitions, indulgence by movement qualities with smooth transitions. Danger and safety may be communicated in much the same way, that is, based on partly the same movement cues inside and outside the body, with secure attachment as the prototype of safety in our early years. Future research will have to inspect these connections, their validity, and the breadth of their implications. 

We generally have the bodily sensitivity to respond to movement rhythms in an attuned way, even though we may in many cases not be explicitly aware of the relevant communicative signals. *On the active side*, we use these signals “intuitively” in embraces or in handling babies, other con­specifics, or animals. We “intuitively” know what to do to address their needs, since we all have communicated our needs in this non- and paraverbal (e.g., vocal) fashion from our earliest days on. All human beings understand the meaning of movement and are experts in its implicit use (decoding and encoding) to a lesser or stronger degree. Even in disorders of the embodied self, where the understanding of these communicative nonverbal signals is sometimes disrupted (e.g., in autism spectrum disorder), much of this understanding is intact and we can use it (e.g., rhythmicity) as resources and as means to relate to our clients on a different level than just the verbal. If we become more conscious of this implicit knowledge and start to use it more in therapy and education, we can develop health, learning, and understanding via a different pathway.

### 4.2. Limitations and Future Directions

Limitations of this study include gender, culture, method, and generalizability issues. Since the study was conducted in a *laboratory setting*, *generalizability* is limited. Many participants rightfully pointed out the artificiality of the embrace situation: the increase in internal validity took its toll in a decrease of external validity. The artificiality of the embrace must be discussed in two respects: (a)In a natural embrace, the sharp phase would not continue for 10 s but just until the point that the embracers actually separate, which is quite soon after the onset of a sharp rhythm. In our study, the drop of the handkerchief happened with a 6–7 s delay in sequence B (where the sharp rhythm was the second rhythm to be applied), whereas it happened with a delay of 3–4 s in sequence A (where the sharp rhythm was the third rhythm applied). This difference may be due to the fact that in Sequence A the embrace had already lasted a lot longer than in Sequence B. We would thus assume that a delay of 3–4 s reflects the more natural conditions of a separation after an indulgent embrace;(b)In natural settings, shape changes almost always accompany the tension-flow changes featured in this study, that is, embracers may retreat their upper bodies or even start to take a step back right after the onset of the sharp rhythm. So in natural conditions, tension-flow and shape- flow go together (they systematically vary together), *while in our experiment only tension-flow was varied in order to isolate the effect of smooth vs. sharp rhythms*. Again, this increased the artificiality of the embrace, increasing internal validity at the expense of decreasing external validity.

*Haptic communication* had been a topic of nonverbal communication research from the 1970s onward (Henley, 1973; Hall, 1990; DePaulo, 1992) [67,79,80]. Gallace and Spence (2010) [59] reviewed how the effects of interpersonal touch are influenced by culture, age, and gender (see also Remland & Jones, 1988) [81].

*Gender aspects.* Gender aspects are probably the most important context factors for a study such as this (Hall, 1990; Hall & Veccia, 1990; Stier & Hall, 1984) [67,82,83]. Hertenstein and Keltner (2011) [84] showed, for example, that males and females differ in their communication of distinct emotions via touch (Nuszbaum et al., 2014) [75]. Yet, in our study, there was no significant gender effect of the person embraced. Men were slightly, but not significantly, more accurate in terms of the main outcome, which is surprising, because studies on nonverbal communication consistently find an advantage for women in the encoding and decoding of nonverbal cues (Hall, 1990; DePaulo, 1992) [67,80]. Then again, men in our study were mainly students of psychology and humanities, where one could hypothesize a higher than average nonverbal sensitivity. Despite the fact that we did not find any gender differences related to the outcomes, we would assume that it creates a difference whether a female confederate embraces a man or a woman, or whether a male confederate embraces a man or a woman. However, for our female embracer, the *sex of the embraced person* and their *“experienced naturalness” of the embrace* were not correlated in this study *r*(1, 60) = −0.046, *p* = 0.728 (*n.s.*). Yet, our preliminary study had clearly shown important gendered pattern of embraces and their phases in natural contexts. A standardized embrace must thus exert different influences on men and women depending on whether the embracer is a man or a woman. The most unnatural embrace in Germany (according to our findings from the preliminary field study) would be a smooth embrace between two men. It would thus not be recommendable to work with a male confederate as the embracer, since according to this rationale in a mixed-sex sample this would introduce considerably more bias than working with a female embracer.

*Culture aspects.* Embraces are highly culturally loaded and highly variable. They have a high context dependency. “For example, in Italy, a hug and kiss on each cheek is considered a common and acceptable form of greeting. By contrast, in Japan the proper greeting consists of a respectful bow and the absence of any tactile contact whatsoever” (Gallace & Spence, 2010; p. 248) [59]. In Germany, people do not as frequently embrace in public business contexts, and do not kiss on the cheek for a greeting or good-bye (as they do in France, Italy, Spain, or the The Netherlands); the regular cultural greeting in Germany is a handshake (see Koch, 2011) [23]. This is a more distant form of greeting and saying good-bye that in many other European countries (Henley, 1973; McDaniel & Andersen, 1998) [60,79]. In this form, you cannot smell or sense the person as much, as when the faces get close for a “kiss” or cheek-to-cheek. Thus, embraces in Germany are more private than in many other European neighboring countries, and in public, they are mostly shared among friends or romantic partners. These context factors must be known and taken into account when interpreting the results. Future studies should extend similar designs to other cultures, or plan cross-cultural comparative studies from the start. While we found two naturalistic studies on observations of airport greetings or goodbyes (of *N* = 152/*N* = 103 pairs of persons) from the US from the 1980s, the aims of these studies were more geared toward the correlation of the length of the greeting to the personal relationship these pairs had (Greenbaum & Rosenfeld, 1980; Heslin & Boss, 1980) [85,86], and thus not targeting our aim. Research on interpersonal touch indicates that the perception of touch also varies greatly—and more than that of other sense modalities—with context, personal preference (such as need for interpersonal touch; Nuszbaum et al., 2010) [68], and perceived intention of the toucher (Schubert et al., 2008; Seibt et al., 2008) [54,55].

*Method aspects.* Furthermore, limitations of the method need to be discussed. Firstly, we used two experimental conditions controlling each other, and thus would suggest to introduce another more neutral/more true control group in the future; secondly, in preparing the data, we replaced eight values of the second embrace with the values of the first embrace, which can be critically discussed: in the second embrace (main trial), nine participants waited to drop their handkerchief for the entire 30 s. We assumed that the discrepancy in the participants’ behavior from the preliminary to the main trial came about, because participants wanted to demonstrate that they are ‘clever/good’ participants by just waiting for the shape-flow signal, that is, the actual stepping back of the embracer in the second embrace. While this was clearly also a sign to end the embrace (movement shape/shape-flow as a cue), it was not the sign we wanted to investigate: our focus was on movement quality/tension-flow and rhythmic transitions as a cue; the applied computational method not only enabled us to keep more persons in the data set, but also showed more of the true effect. As displayed in Appendix B, the analysis results in significant findings, no matter whether this replacement is done or not, with roughly the same effect sizes in all three computational ways possible (see Appendix B). Moreover, as described above, the sharp rhythm should not last for 10 s, since in natural embraces the sharp phase is shorter than the smooth phase. Future studies could refine digital time measurements to reach higher accuracy of reaction times, yet, the time measurement was of sufficient accuracy for the purpose of this study.

### 4.3. Conclusions and Implications for Future Research

In sum, the findings of this study support the notion that body rhythms are employed and understood as mostly implicit signals communicating specific meanings, such as the wish to separate from an embrace. Such rhythmic behaviors and their meaning are fundamental elements of the communication with caregivers from the beginning of life, and with partners and friends later in life. Infants seem to be highly sensitive to the dynamic nuances, demonstrating a high plasticity at an early age (Shai & Belsky, 2011a; 2011b; Trevarthen, 1999) [29,63,64] flexibly adapting to the family and cultural context. Well known in the area of music therapy, Trevarthen (1999) [29] has termed this same basic rhythmicity *“musicality”*, and—based on Bruner’s essential theories on culture (1990) [87]—puts it this way: “*the parameters of musicality are intrinsically determined in the brain, or innate, and necessary for human development. Through their effects in emotional integration and the collaborative learning that leads to mastery of cultural knowledge, cultural skills, and language, they express the essential generator of human cognitive development*” (Trevarthen, 1999; p. 155) [29]. Adaptive mechanisms regarding dynamic communicative patterns (rhythmicity/musicality) are assumed to lay the foundation of our culture: family culture, subculture, and larger culture and likely also play a role in interpersonal attraction as well as in aesthetic experience (e.g., of nature or of the arts).

The connection between *resonance* and *aesthetics* is made by Freedberg and Gallese (2007) [88]. The authors challenge the primacy of cognition in responses to art, and propose that an important part of the aesthetic response consists of the activation of embodied mechanisms including the simulation of actions, emotions, and corporeal sensation (see also Koch & Fuchs, 2011; Koch 2017; Niedenthal et al., 2005) [89,90,91]. The mechanism of motor simulation, coupled with the emotional resonance it triggers, as suggested by Lipps (1910) [92], is likely to be a crucial component of the aesthetic experience of objects in art works: even a still-life can be ‘animated’ by the embodied simulation it evokes in the observer… (Freedberg & Gallese, 2007, p. 201) [88]. On a kinetic-kinaesthetic and a tactile-haptic level, we may have aesthetic experiences with habitual rhythmicity that we are used to and have been exposed to in pleasant situations. We may find those patterns more beautiful than others, and alternations in habitual patterns more interesting and understandable than completely different patterns. In a video-clip of a demented old lady, she was gently stroked from the top of the nose to the end of the cheek in a certain (swaying) rhythm: her entire face relaxed and she was getting calm and comfortable, crying some tears of being moved, and then smiling with bliss. She could relate this way of being touched to how she was stroked by her mother, when she was a child, and a relation with beauty clearly stood out. Future research is called to investigate the connection between such bodily resonance, aesthetics, memory, and emotion.

The present study provides evidence supporting the meaning of (micro-)movements in personal and interpersonal communication that future research can build upon. For cognitive scientists, it underlines the utility of accessing the knowledge of movement analysis for a more differentiated theory and method of bodily systems that influence embodiment effects in a systematic way, and for movement analysts, it emphasizes the integration and servicing of their knowledge for other areas of research. Following *embodiment* theory, our bodies influence the ways we think and feel in multiple important respects (Gallagher, 2006; Gibbs, 2005; Johnson, 2008; Pfeifer & Bongard, 2006) [15,16,17,93]. Putting more effort into the research of this knowledge and its implications can advance our understanding of communication, expression, human development, human relations, emotion regulation, and clinical recovery.

## Figures and Tables

**Figure 1 behavsci-07-00080-f001:**
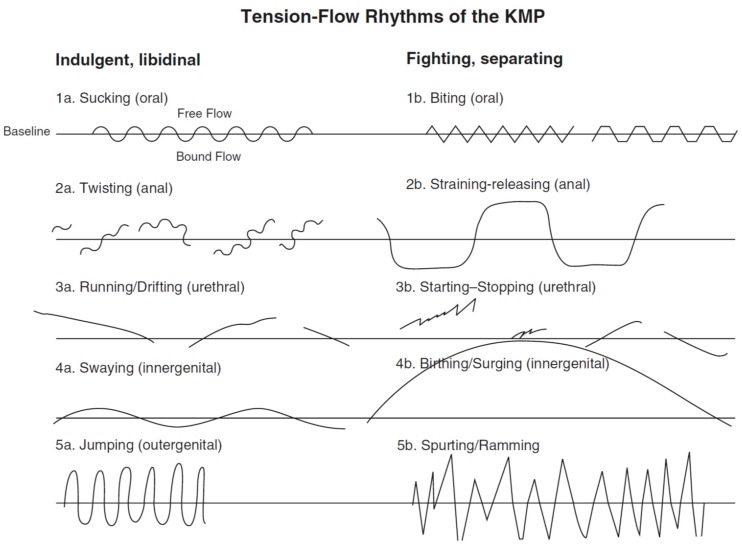
Prototypic Tension-Flow Rhythms of the Kestenberg Movement Profile (KMP; Kestenberg, 1975/95) [2]. Note: Tension-flow rhythms are the rhythmic changes between tension and relaxation in the body of an individual; they are notated on paper, yielding a continuous tension-flow line on a timeline from left to right; downward writing indicates moving into higher tension; upward into lower tension (by convention); they are similar to Stern’s concept of vitality affects (2012) [45], but more on the active side (and thus more directly connected to control, self-efficacy, and self-regulation); in the embrace study, we used the sucking, swaying, and biting rhythm (figure adapted from Koch & Sossin, 2012) [5].

**Figure 2 behavsci-07-00080-f002:**
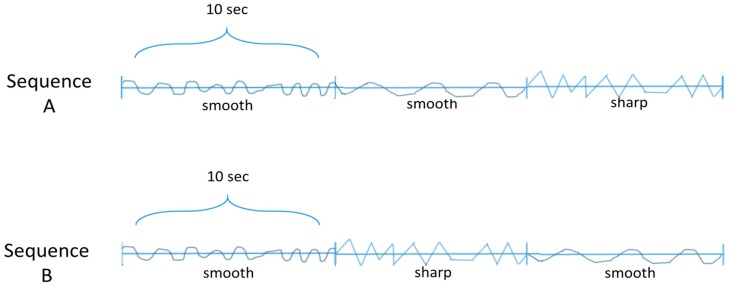
The two embrace conditions (the first rhythm was a sucking rhythm in both sequences, the second either a swaying (**A**) or a biting rhythm (**B**); and the last either a biting (**A**) or a swaying-rhythm (**B**); the biting rhythm was employed, because it was the most natural sharp rhythm in embraces, according to our naturalistic pre-study); each sequence lasted 30 s and consisted of three rhythms; each phase lasted 10 s.

**Figure 3 behavsci-07-00080-f003:**
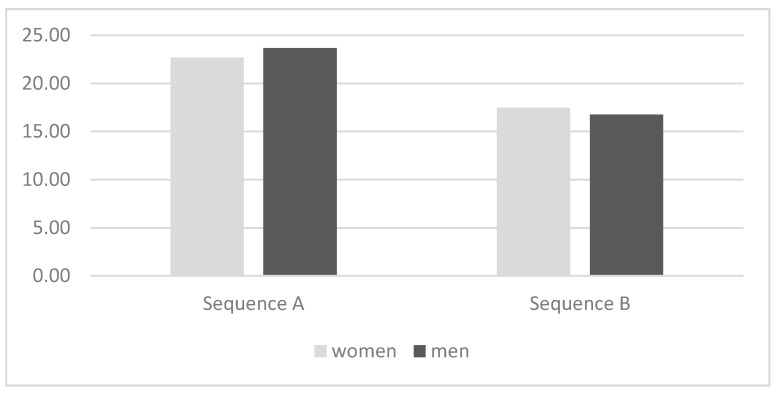
Times from Onset of Embrace to Signal Detection by Gender (*N* = 60). Note. Sequence A: smoot-smoot-sharp; Sequence B: smooth-sharp-smooth; the difference for time of signal was significant with *F*(1, 60) = 18.90, *p* = 0.000, *eta*^2^ = 0.25; the difference in time of signal between men *M* = 20.21 (*SD* = 5.84) and women *M* = 20.29 (*SD* = 6.16) was not significant, *F*(1, 59) = 0.07, *p* = 0.789, *eta*^2^ = 0.00.

**Table 1 behavsci-07-00080-t001:** Descriptive Statistics of Time of Signal, Phase of Drop, and Affect Change (*N* = 60).

	Sequence	Mean	SD	N
**Time of Signal**	A: sm-sm-sh	23.13	4.85	31
	B: sm-sh-sm	17.17	5.72	29
**Phase of Signal**	A: sm-sm-sh	2.74	0.51	31
	B: sm-sh-sm	2.24	0.43	29
**Affect Change**	A: sm-sm-sh	0.28	0.52	31
	B: sm-sh-sm	0.28	0.57	29

Note: SD = Standard Deviation; *N* = Number of participants; Time of Signal: time measured from onset of embrace to drop of the handkerchief; Phase of Signal: 1 = Phase 1 (smooth rhythm), 2 = Phase 2 (smooth in Sequence A; sharp in Sequence B); 3 = Phase 3 (sharp in Sequence A, smooth in Sequence B), means low, because of error drops (e.g., in Phase 1); Affect Change = Posttest-Pretest on MBAS-Scale (Koch & Müller, 2007) [72]; sm = smooth rhythm; sh = sharp rhythm; Degrees of freedom = 1.

**Table 2 behavsci-07-00080-t002:** Crosstab for Phase of Signal.

		Rhythms Sequence		
		Sequence A	Sequence B	Total
Phase of Signal	Phase 2 (s 10–20)	6	22	28
	Phase 3 (s 20–30)	24	7	31
Total		30	29	59

Note: There was one participant in both trials that already signaled in the first phase and one participant that signaled only after the last phase; these two participants (both in the condition Sequence A) were taken out of the analysis, resulting in *N* = 59 (30 vs. 29), from these 46 persons (78%) understood the signal, and 13 did not, according to the results for *phase of signal*.

**Table 3 behavsci-07-00080-t003:** Inferential Statistics of Time of Signal (Drop of Handkerchief) and Affect Change (*N* = 60).

	Sequence	*F*	*P*	*eta*^2^
**Time of Signal (drop)**	A: sm-sm-sh	18.90	0.000	0.25
	B: sm-sh-sm			
**Affect Change**	A: sm-sm-sh	0.007	0.934	0.00
	B: sm-sh-sm			

Note: SD = Standard Deviation; N = Number of participants; Time of Signal: measured from onset of embrace to drop of the handkerchief; Affect Change = Posttest-Pretest on MBAS-Scale (Koch & Müller, 2007) [72]; sm = smooth rhythm; sh = sharp rhythm; Degrees of freedom = 1.

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
