# Peer review of "Psychology of the Embrace: How Body Rhythms Communicate the Need to Indulge or Separate"

_behavsci, 2017, doi:10.3390/bs7040080_

Round 1

Reviewer 1 Report

This is an thought- provoking and interesting article providing a detailed study about a topic that is of relevance to dance movement therapists and other like professionals.

However, several issues arise for me in the reading of it, which if repaired would make the article publishable at a good standard

Abstract:

For me, this abstract goes too quickly to too much detail, without providing the reader with any background as to what the study is about in a general sense and what the research question is. Jargon terms such as smooth, indulging rhythms are not explained, nor the theory that underpins them. The final sentences provide a bit more context but not sufficient for this reader to be convinced why the study might be useful.

English expression:

this article would be improved with greater attention to English expression and grammar. Some examples: 

116: This half of the sentence is confusing: These known effects can be sorted into the system of movement analysis

134; rhythm can help to re-gain experiences of self-efficacy SOMETHING MISSING HERE a crucial component of psychological health (Bandura, 1977). 135 

162: By relying on KMP-theory, the present embrace study again succeeds in separating both aspects, which allows US? to investigate the dimension of movement quality (smooth vs. 162 sharp rhythms) independently of the dimension of movement shape regarding their affective and 163 communicative effects. 164 

309: is mayor meant to be major

348: irritated

466: lose not loose

504: true vs untrue”, missing commas

722: rationale not rational

782: easthetic should be aesthetic

Concept explication:

89/90    I am unclear what ‘approach or avoidance movements’ are and no definition is provided as they are discussed.  

92: Affect and attitudes are similarly undefined

122: interoception from exteroception 

641: This paragraph is a very clear explanation about the study and its function.

Article structure:

This is the area where I think the article is weakest- improvement needed about when information appears

167: This intro to Kestenberg rhythms should come much earlier- this frames the whole discussion but I read four pages before I find out what the theory is

245: the research question does not formally appear until page 6, buried in a paragraph that seems not structurally different than those coming before. What about a heading saying ‘research question’?  Then the lit review seems to continue, followed at 274 by further explication of method, followed by more lit review, then at 286 more method.

328: The aim of the study is offered at 328, page 7 of the article, without any heading to signpost it.

702: This section seems like lit review. Im not sure why its there in limitations.

725: this section seems to be re-introducing literature and repeating previously discussed research

777: this feels like a whole lot of new literature and ideas being introduced that were not previously mentioned

786: this entire story seems unconnected to the point of the article and would be better either deleted or included in literature and referenced in discussion or future research. Neither is it referenced

794: is where we hear for the first time what the implications of the study are.  This should come right at the beginning, helping the reader work out what use the study might be.

Methodological issues

There are several issues with the method that should be improved and would make it a better article.

161: I’m not sure if this is a question of expression, or that the authors are claiming that their study is successful before they have reported it, but this sentence stands out for me as an inappropriate claim at this point: 161 ‘the present embrace study again succeeds in separating both aspects’. Does the present study address this challenge? Why is it ‘again succeeds,’ when you haven’t yet told me what you are doing or why?

345: I think that would be method (how this was signalled) rather than hypothesis

349: ‘The drop of the handkerchief was in many cases not noticed by the embracer, in others at least not noticed for sure’ 

Wouldn’t this be a methodological issue to report in findings, not in hypothesis?

427:' A preliminary test of the experimental set-up had shown that for many participants it was important to know the sex of the embracing person'.

This preliminary test should be explained properly – what was tested and how was it tested?

471: 'Participants probably tried to be smart the second time of the embrace' 

 I think this is an unsubstantiated comment and should be deleted unless you can offer some justification

Author Response

Dear Reviewer 1: 

I have changed almost everything you had reccommended, except a few things, that I would like to ennumerate here (your comments in italics): 

167: This intro to Kestenberg rhythms should come much earlier- this frames the whole discussion but I read four pages before I find out what the theory is

--> after moving it around several times, I decided to leave it where it is, since other aspects then started to seem not in the correct order... Instead I inserted a comment right on the first page that not only leads to videos for a better understanding of the operationalization, but also directly to the KMP-theory starting at 1.3

245: the research question does not formally appear until page 6, buried in a paragraph that seems not structurally different than those coming before. What about a heading saying ‘research question’?  Then the lit review seems to continue, followed at 274 by further explication of method, followed by more lit review, then at 286 more method. 

345: I think that would be method (how this was signalled) rather than hypothesis à it is a method hypothesis (added and also changed the heading to Hypothesis and Operationalization)

à the research question is now directly stated in the abstract, I hope that satisfies the need for having it introduced earlier and more clearly.

702: This section seems like lit review. Im not sure why its there in limitations. 

--> Because it is the review related to the two areas that needed to be discussed as posing two important limitations: gender and culture aspects.

All other comments were adressed (see attached manuscript), we hope to your satisfaction. 

Thank you so much for your valuable comments. They made the manuscript better to read, more logical and more comprehensive. 

Best, 

Sabine Koch

Reviewer 2 Report

A behavioral study on the nonverbal dynamics of signaling the wish to continue or to stop a current embrace are studied. A confederate hugged a participant and expressed, in well-defined intervals determined by the experimental protocol, either smooth or sharp movement rhythms. The participants was asked to signal the moment at which s/he thought the confederate would like to stop the embrace. It was found that sharp compared to smooth movements were used by the participants as intuitive cues for stopping the embrace.

This is a fascinating, highly original, and also entertaining study that nevertheless provides theoretically interesting and important insights. The literature review (especially incorporating the classic kiki-bouba effect) is splendid, the methods are rigorous, and the reporting of the results (although a bit verbose and laborious) is proper. I enjoyed reading this interesting piece that will be of great attraction to the readership of Behavioral Sciences. I only have these minor suggestions to further improve this already excellent piece.

In the method section of the main experiment (not in the preliminary studies) the authors should describe in more detail what actually the smooth and sharp rhythms were. I see that they introduced this concept extensively in the intro, but we need the exact operationalization also in the methods. Ideally (but not necessarily), the authors upload a video of an exemplary hug to youtube and add the link to this movie to the manuscript.

I do not see why for this simply between-subjects comparison between two means an F-statistic (a whole ANOVA) is necessary. Usually, for such a comparison one uses a simple independent samples t-test. I know this is an old discussion, since the t-test will have the same significance level as the two-level ANOVA, but readers are simply more used to a t-test when only two means are compared to each other. I see that the authors do not an ANOVA to insert their covariates (rendering it an ANCOVA), but I suggest simply also adding the t-test to provide a more generally known and comparable effect size (namely Cohen’s d). Partial eta is of course an effect size, but it is harder to interpret than Cohen’s d. For d there are generally known standards (what is a small, medium, or large effect).

Thank you for this delightful read,

Sascha Topolinski

Author Response

Dear Sascha, 

Thank you so much for your valuable and positive feedback. :-)

This means a lot to me.

I have now offered Behavior Science to either upload an example of the original video footage (if I can still get informed consent for its use), or to record an example just for this purpose, or to integrate the link to the KMP website that contains video examples of all of the rhythms. Thank you for this splendit idea! 

Regarding your request for t-tests: When I tried to do this, I noticed that I had in fact computed the ANCOVA to be able to integrate both covariates (sex and NFIT) into the main analysis, which I cannot do if I compute t-Tests. Plus, I get a problem with multiple testing, if I do t-tests. 

Thus, I would like to keep the ANCOVA, but I additionally reported Cohen's d's, and hope that this satisfies your request. If not, please get back to me on your idea with more details, so I am sure I correctly understand it.  

Best, Sabine